# Convolutional Long-Short Term Memory Network with Multi-Head Attention Mechanism for Traffic Flow Prediction

**DOI:** 10.3390/s22207994

**Published:** 2022-10-20

**Authors:** Yupeng Wei, Hongrui Liu

**Affiliations:** Department of Industrial and Systems Engineering, San Jose State University, San Jose, CA 95192, USA

**Keywords:** traffic flow prediction, deep learning, convolutional LSTM, attention mechanism

## Abstract

Accurate predictive modeling of traffic flow is critically important as it allows transportation users to make wise decisions to circumvent traffic congestion regions. The advanced development of sensing technology makes big data more affordable and accessible, meaning that data-driven methods have been increasingly adopted for traffic flow prediction. Although numerous data-driven methods have been introduced for traffic flow predictions, existing data-driven methods cannot consider the correlation of the extracted high-dimensional features and cannot use the most relevant part of the traffic flow data to make predictions. To address these issues, this work proposes a decoder convolutional LSTM network, where the convolutional operation is used to consider the correlation of the high-dimensional features, and the LSTM network is used to consider the temporal correlation of traffic flow data. Moreover, the multi-head attention mechanism is introduced to use the most relevant portion of the traffic data to make predictions so that the prediction performance can be improved. A traffic flow dataset collected from the Caltrans Performance Measurement System (PeMS) database is used to demonstrate the effectiveness of the proposed method.

## 1. Introduction

Traffic congestion results in reduced efficiency of transportation infrastructure, increased traveling time, and a waste of energy fuel [1,2,3]. According to a report by Nationwide, 1.9 billion gallons of fuel are wasted every year as a result of traffic congestion [4]. Traffic congestion could be induced by numerous factors, such as bottlenecks, traffic accidents, and severe weather conditions. To address the issue of traffic congestion, traffic flow prediction has gained much attention in the recent decade. Accurate predictive modeling of traffic flow is critically important as it allows transportation users to make wise decisions to circumvent traffic congestion regions [5]. Therefore, commuter and shipment activities could be effectively scheduled to increase moving efficiency. Moreover, accurate predictive modeling of traffic flow can also assist in reducing carbon emissions and traffic incident possibilities.

The advanced development of sensing technology makes big data more affordable and accessible, and thus, data-driven methods have been increasingly adopted for the predictive modeling of traffic flow. Data-driven methods can be classified into two categories: machine learning methods and deep learning methods [6,7,8,9,10]. In comparison with machine learning methods, deep learning methods have gained more attention from both academia and industry in traffic flow predictions due to their extraordinary prediction fidelity and robustness. Among these deep learning methods, artificial neural networks (ANNs) and autoencoder-based methods have been widely used for traffic flow predictions as these methods are capable of decomposing the original traffic flow data into features located at a higher dimensional feature space, and these high-dimensional features can reveal the latent information in the traffic flow data. However, there are two primary issues for ANNs and autoencoders: (1) they can not take the temporal correlation of traffic flow data into account; (2) they can not consider the correlation of the extracted high-dimensional features. To consider the temporal correlation of traffic flow data, deep learning methods with recurrent characteristics are adopted, such as long short-term memory (LSTM), recurrent neural network (RNN), and gated recurrent unit (GRU). While these deep learning methods with recurrent characteristics are promising, they are not able to use the most relevant part of the traffic flow data to make predictions, which leads to a higher prediction time and a worse prediction accuracy. To address these issues, this work introduces a novel deep learning-based framework to consider the temporal correlation of traffic flow data, the correlation of the extracted high-dimensional features, and the most relevant part of the traffic flow data to make predictions in a unified manner. More specifically, a decoder network is firstly proposed to decompose the traffic flow data into high-dimensional features. Second, a convolutional LSTM network is introduced to simultaneously consider the correlation of the decomposed high-dimensional features and the temporal correlation of traffic flow data, where the convolutional operation is used to consider the correlation of the high-dimensional features, and the LSTM network is used to consider the temporal correlation of traffic flow data. Next, the multi-head attention mechanism is introduced to use the most relevant portion of the traffic data to make predictions so that the prediction performance can be improved. The primary contribution of this work can be summarized as follows:A decoder network is introduced to decompose the original traffic flow data into features located at a higher-dimensional feature space.A convolutional LSTM network is introduced to consider the correlation of the high dimensional features and the temporal correlation of traffic flow data.A multi-head attention mechanism is introduced to use the most relevant portion of the traffic data to make predictions so that the prediction performance can be improved.

The remainder of this paper is organized as follows. Section 2 reviews data-driven methods reported in the literature for traffic flow predictions. Section 3 introduces the proposed deep learning model. Section 4 demonstrates the effectiveness of the proposed method utilizing the traffic flow data from the Caltrans Performance Measurement System (PeMS) database. Section 5 concludes this research work and directs future work.

## 2. Literature Review

In the context of traffic flow predictions, data-driven methods can be classified into two categories: machine learning [11,12,13] and deep learning methods [14,15]. These machine learning methods include support vector regression [16], random forest [17], Gaussian process [18], Bayesian models [19], and so on. For example, Tang et al. [20] combined the support vector machine method with multiple denoising mechanisms to predict the traffic flow. A dataset collected by the real-time detectors located in the city of Minneapolis was used to evaluate the performance of the proposed methods. The simulation results have shown that the denoising mechanisms could boost the performance of the support vector machine. Zhang et al. [21] introduced a hybrid framework based upon support vector regression to predict the traffic flow, where the random forest method was implemented for feature selections, and the genetic algorithm was adopted to determine the model hyperparameters. The simulation results have shown that the proposed methodology enables better prediction accuracy. Xu et al. [22] introduced a scalable Gaussian process model for large-scale traffic flow predictions. The proposed model combined the Gaussian process with alternative directional methods for paralleling and optimizing hyperparameters during the training process. Wang et al. [23] presented a vicinity Gaussian process method for short-term traffic flow prediction under the conditions of missing data with measuring errors. In the proposed model, a directed graph was constructed based on the traffic network, a dissimilarity matrix and a proper cost function were selected to boost the prediction performance. Zhu et al. [24] introduced a linear conditional Gaussian process method, where temporal and spatial correlations of traffic flow were taken into account. A simulated traffic dataset was adopted to evaluate the effectiveness of the Gaussian process method, and simulation results have shown that the utilization of both spatial and temporal data can dramatically boost prediction accuracy. Li et al. [25] presented a Bayesian network to tackle the node selection challenge in traffic flow prediction. Experimental results have shown that the proposed directed correlation-based Bayesian network method results in a sparse model and better performance in traffic flow prediction.

With the advanced improvement of computational power, deep learning methods are increasingly adopted in traffic flow prediction due to their extraordinary performance. These deep learning methods include LSTM [26,27], gated recurrent neural network (GRU) [28,29], recurrent neural network (RNN) [30,31], graph neural network (GNN) [32,33,34], and so on. For instance, Tian et al. [35] introduced LSTM-based predictive modeling of traffic flow, where a smoothing function was implemented to deal with the missing data points, and the LSTM was used to capture the prediction residual. Two traffic flow datasets were used to evaluate the performance of the proposed methodology, and the results have shown that the smoothing function can boost the performance of the predictive model. Dai et al. [36] integrated the spatial-temporal analysis with a GRU network to forecast the traffic flow in a short time interval. In the proposed method, the GRU model was applied to process the spatial-temporal features extracted from the collected traffic data. The simulation results have shown that the GRU outperforms the convolutional neural network (CNN) in both prediction accuracy and robustness. Zhene et al. [37] combined the CNN with RNN for urban traffic flow predictions, where CNN was adopted to extract attributes from traffic flow data and RNN was implemented to make predictions. In comparison with the traditional RNN, the proposed RNN was able to process multiple temporal features simultaneously. The experimental results have demonstrated that online traffic flow prediction could be achieved with high precision by using the proposed methodology. Luo et al. [38] introduced a k-nearest neighbor-based (KNN) LSTM method to extract temporal and spatial correlations, where KNN was utilized to capture spatial correlations and LSTM was adopted to further extract temporal correlations. A dataset provided by the University of Minnesota Duluth Data center was utilized to demonstrate the effectiveness of the proposed methods, and the results have indicated that the proposed method outperforms the auto-regressive integrated moving average and wavelet neural network in terms of prediction accuracy. Zhu et al. [39] integrated the GNN with RNN to extract the spatial and temporal correlations of traffic data. The belief rule-based algorithm was adopted for data fusion, and the fused traffic data were fed into the proposed methodology for traffic flow prediction. Yu et al. [40] presented a novel GNN methodology to predict the traffic flow, in which a weighted undirected graph was utilized to differentiate the density of connected roads. A simulation model was introduced to simulate the traffic propagation, and the simulation results were considered in the GNN model for online traffic flow prediction. The simulation results have shown that the proposed GNN outperforms the traditional GNN in traffic flow predictions. More details about applying GNN for traffic flow predictions can be found in [41].

While numerous data-driven methods have been studied to predict traffic flow under various conditions, some issues still exist with these methods. The existing data-driven methods can not consider the correlation of the extracted high-dimensional features and can not use the most relevant part of the traffic flow data to make predictions, which leads to a higher prediction time and a worse prediction accuracy. To deal with these issues, this work proposes a decoder convolutional LSTM network to simultaneously consider the correlation of the decomposed high-dimensional features and the temporal correlation of traffic flow data, where the convolutional operation is used to consider the correlation of the high-dimensional features, and the LSTM network is used to consider the temporal correlation of traffic flow data. Moreover, a multi-head attention mechanism is introduced to use the most relevant portion of the traffic data to make predictions so that the prediction performance can be improved.

## 3. Convolutional LSTM with Multi-Head Attention Mechanism

This section introduces the convolutional LSTM with a multi-head attention mechanism. Figure 1 shows the framework of the proposed deep learning approach. First, a moving window with a fixed window size is utilized to split raw traffic flow into historical traffic flow as features and future traffic flow as labels. The historical traffic flow is fed into a decoder network to be decomposed into multiple time-series signals. The decomposed signals are fed into the convolutional LSTM network to consider the correlation of the decomposed high dimensional features and the temporal correlation of traffic flow data. The outputs of the convolutional LSTM are transited to the multi-head attention model for traffic flow prediction. Next, the prediction loss is calculated based on the future traffic flow and predicted traffic flow, and the backpropagation algorithm is adopted to train the proposed method. More details of the proposed deep learning approach are provided in the following subsections.

### 3.1. Decoder Network for Traffic Data Decomposition

The initial step of the proposed method is to decompose the traffic flow so that the most useful latent information can be reflected and the data can be better analyzed. To decompose the traffic flow data, this research uses a decoder network that stacks multiple fully connected layers. The output of the decoder network can be written as Equation (Equation 1),
(1)$$D_{i,L}=f_{L}\ldots \left[f_{l}\ldots \left[f_{2}\left[f_{1}\left(X_{i}\right)\right]\right]\right]$$
where $X_{i}\in R^{1\times T}$ represent the traffic flow data for data sample *i*; *L* refers to the total number of stacked fully connected layers in the decoder network; $D_{i,L}\in R^{m\times T}$ refers to the output of the decoder network for data sample *i*; *m* represents the number of hidden nodes in the fully connected layers of the decoder network; *T* represents the length of the historical traffic flow; and $f_{l}\left(\cdot \right)$ can be given by Equation (Equation 2).
(2)$$f_{l}\left(\cdot \right):=Relu\left(W_{l}\cdot D_{i,l-1}+b_{l}\right)$$

In Equation (Equation 2), Relu represents the rectified linear unit activation function; $W_{l}$ refers to the kernel weight matrix at the *l*-th fully connected layer in the decoder network; $D_{i,l-1}$ represents the output of the l−1-th fully connected layer for data sample *i*; and $b_{l}$ represents the bias weight matrix at the *l*-th fully connected layer. Next, the output $D_{i,L}$ of the decoder network is fed into the convolutional LSTM network to consider the correlation of the decomposed high-dimensional features and the temporal correlation of traffic flow data.

### 3.2. Convolutional LSTM Cell

The traditional LSTM is capable of considering the temporal correlation of traffic flow data. However, the traditional LSTM fails to consider the correlation of the decomposed high-dimensional features. To address this issue, this research aims to introduce the convolutional LSTM cell that incorporates a convolutional operation into the traditional LSTM cell so that both the temporal correlation of traffic flow data and the correlation of the decomposed high-dimensional features can be considered in a unified manner [42]. Figure 2 shows the framework of the convolutional LSTM cell. In the convolutional LSTM cell, the output vector $d_{i,L}^{\left(t\right)}$ of the decoder network at time *t* and the hidden state $h_{i,t-1}$ of the one-dimensional convolutional LSTM cell at the prior time point t−1 are fed into the one-dimensional convolutional LSTM cell to perform the weighted convolutional operations. Such convolutional operations can consider the correlation of the decomposed high dimensional features $D_{i,L}$. The recurrent usage of the convolutional LSTM cell can extract temporal correlations, and the output of this cell can be written as Equation (Equation 3),
(3)$$\begin{array}{cc} f_{i,t} & =\sigma (C_{i,f}+W_{f,c}\circ c_{i,t-1}+b_{f})\\ a_{i,t} & =\sigma (C_{i,a}+W_{a,c}\circ c_{i,t-1}+b_{a})\\ c_{i,t} & =f_{i,t}\circ c_{i,t-1}+a_{i,t}\circ Tanh\left(C_{i,c}+b_{c}\right)\\ o_{i,t} & =\sigma (C_{i,o}+W_{o,c}\circ c_{i,t}+b_{o})\\ h_{i,t} & =o_{i,t}\circ \sigma \left(c_{i,t}\right) \end{array}$$
where $f_{i,t},a_{i,t},c_{i,t},o_{i,t}$, respectively, refer to the outputs of the forget gate, input gate, memory cell, and output gate; $W_{f,c},W_{a,c},W_{o,c}$ represent the trainable matrices for the forget gate, input gate, and output gate, respectively; $b_{f},b_{a},b_{c},b_{o}$ represent the bias vectors for the forget gate, input gate, memory cell, and output gate; σ refers to the sigmoid function; Tanh refers to the hyperbolic tangent function.

Moreover, $C_{i,f},C_{i,a},C_{i,c},C_{i,o}$, respectively, refer to the outputs of the convolutional operations at the forget gate, input gate, memory cell, and output gate. These convolutional outputs can be written as Equation (Equation 4), where ∗ refers to the convolutional multiplication; $W_{f,d}$ and $W_{f,h}$ refer to the kernel matrices of the convolutional operations at the forget gate; $W_{a,d}$ and $W_{a,h}$ are the kernel matrices of the convolutional operations at the input gate; $W_{c,d}$ and $W_{c,h}$ represent the kernel matrices of the convolutional operations in the memory cell; and $W_{o,d}$ and $W_{o,h}$ represent the kernel matrices of the convolutional operations at the output gate.
(4)$$\left\{\begin{array}{c} C_{i,f}=W_{f,d}*d_{i,L}^{\left(t\right)}+W_{f,h}*h_{i,t-1}\\ C_{i,a}=W_{a,d}*d_{i,L}^{\left(t\right)}+W_{a,h}*h_{i,t-1}\\ C_{i,c}=W_{c,d}*d_{i,L}^{\left(t\right)}+W_{c,h}*h_{i,t-1}\\ C_{i,o}=W_{o,d}*d_{i,L}^{\left(t\right)}+W_{o,h}*h_{i,t-1} \end{array}\right.$$

In summary, the convolutional LSTM cell integrates the convolutional operations with the traditional LSTM cell, where the convolutional operations are adopted to consider the correlation of the decomposed high-dimensional features $D_{i,L}$ and the traditional LSTM cell is utilized to extract the temporal correlations of traffic flow data. The integration of the convolutional operation with the traditional LSTM cell allows the neural network to consider both the correlation of the decomposed high-dimensional features and the temporal correlation of traffic flow data. Next, the hidden outputs, $h_{i,t}$ for all *t*, of the convolutional LSTM cell are fed into the multi-head attention mechanism for the final prediction.

### 3.3. Multi-Head Attention Model

In the recent decade, the attention mechanism [43,44] has been introduced to deal with time series as it is capable of using the most relevant proportion of a time series to make predictions. The primary theory of the attention mechanism is simulating the data retrieval process in the data management system. To retrieve data, a query should be inserted into a data management system. If the query is matched with a key, the value associated with the key will be retrieved. Equation (Equation 5) shows the construction process of queries $Q_{i}$, keys $K_{i}$, and values $V_{i}$ for traffic flow predictions.
(5)$$\left(W_{Q},W_{K},W_{V}\right)\cdot H_{i}=\left(Q_{i},K_{i},V_{i}\right)$$

In Equation (Equation 5), $H_{i}$ represents the hidden outputs of the convolutional LSTM network for data sample *i*, and $H_{i}$ can be written as $H_{i}=\left(h_{i,1},\cdots ,h_{i,t},\cdots ,h_{i,T}\right)$; and $W_{Q}\in R^{r\times T},W_{K}\in R^{r\times T},W_{V}\in R^{r\times T}$ are trainable weight matrices. To use the most relevant portion of the values *V*, the attention vector a should be obtained by using Equation (Equation 6), where SoftMax is the normalized exponential function.
(6)$$a=SoftMax(Q_{i}\cdot K_{i}^{\prime }/\sqrt{T})$$

To retrieve the most relevant part of the values *V*, the attention vector is multiplied by the value matrix, which can be written as $O_{i}=aV_{i}$.

The multi-head attention mechanism stacks the multiple attention model [45,46]. Figure 3 presents the framework of the multi-head attention model for traffic flow prediction. The attention vector of the multi-head attention mechanism can be written as $a_{h}=SoftMax\left(W_{Q}^{\left(h\right)}H_{i}\cdot {\left(W_{K}^{\left(h\right)}H_{i}\right)}^{\prime }/\sqrt{T}\right)$, where $W_{Q}^{\left(h\right)},W_{K}^{\left(h\right)},W_{V}^{\left(h\right)}$ are trainable weight matrices of the *h*-th attention model; and $a_{h}$ is the attention vector of the *h*-th attention model. The output of the *h*-th attention model is written as $O_{i,h}=a_{h}\left(W_{V}^{\left(h\right)}H_{i}\right)$.

Next, the output of all attention models is concatenated, which can be written as Equation (Equation 7), where *H* is the number of attention models and has been stacked in the multi-head attention model.
(7)$$C_{i}=Concat\left\{O_{i,1},\cdots ,O_{i,h},\cdots ,O_{i,H}\right\}$$

Next, the concatenated output C is fed into a fully connected layer for final predictions. The training loss of the traffic flow prediction is written as Equation (Equation 8), where *N* refers to the total amount of data samples; $y_{i,j}$ is the true traffic flow for sample *i* at time *j*; and ${\hat{y}}_{i,j}$ is the predicted traffic flow for sample *i* at time *j*.
(8)$$L=\frac{1}{N\times T}\sum_{i=1}^{N}\sum_{j=1}^{T}{\left(y_{i,j}-{\hat{y}}_{i,j}\right)}^{2}$$

The backpropagation algorithm is utilized for training the proposed deep learning model. Table 1 presents the training process of the proposed method. First, the weight matrices in the deep learning model are randomly initialized, the traffic flow data and labels are prepared, and the learning rate is initialized. Next, the traffic flow data $X_{i}$ for data sample *i* are fed into the decoder network to decompose the traffic flow data into multiple parts. The output $D_{i,L}$ of the decoder network is fed into the convolutional LSTM layer to extract temporal and spatial correlations, and the output of this layer is $H_{i}$. Next, $H_{i}$ is fed into the multi-head attention model to use the most relevant portion of the features extracted by the convolutional LSTM layer. The output of the multi-head attention model $C_{i}$ is fed into the fully connected layers for traffic flow predictions, and the trainable weight matrices are updated in each training iteration.

## 4. Case Study

In this section, a real-world traffic flow dataset was used to demonstrate the effectiveness of the proposed deep learning approach. The following subsections provide dataset descriptions, evaluation metrics, model architecture, and prediction results.

### 4.1. Dataset Description

Traffic flow data collected by the Caltrans Performance Measurement System (PeMS) was utilized to demonstrate the effectiveness of the proposed methodology. The dataset was collected in real-time from over 40,000 unique detectors located on the freeway in the state of California [47]. The collected dataset aggregated hourly traffic flow data obtained from the corresponding detection station. In this study, we used two cases to demonstrate the effectiveness of the proposed method. The first case used the traffic flow data collected from January to March in the year 2022 located at the I5-North freeway, where the post-mile range is from 495.73 to 621.42 in the state of California. The second case used the traffic flow data collected from February to April in the year 2022 located at the I5-North freeway, where the post-mile range is from 495.73 to 621.42 in the state of California. The post-mile refers to the range of routes that move through individual counties in the state of California. For both two cases, the data for the first two months were used to train the proposed deep learning model, and the remaining month was used to test the proposed model. Figure 4 highlights the range of the post-mile 495.73 to 621.42 at the freeway I5-North. To avoid loss of generality, both training and test data were standardized. In this work, we use the data rescaling method to standardize all data to guarantee that both vehicle miles traveled (VMT) and vehicle hours traveled (VHT) are on the same scale. The data rescaling method refers to multiplying each data point by a constant factor, where the factors for VMT and VHT are $10^{-5}$ and $10^{-3}$, respectively.

### 4.2. Evaluation Metric

To evaluate the performance of the proposed methodology, this study adopts the root mean squared error (RMSE) and mean absolute error (MAE). The RMSE and MAE can be defined by using Equation (Equation 9), where *N* is the total amount of data samples; $y_{i,j}$ refers to the true traffic flow for the sample *i* at time *j*; and ${\hat{y}}_{i,j}$ represents the predicted traffic flow for the sample *i* at time *j*.
(9)$$\begin{array}{cc} \; & RMSE={\left(\frac{1}{N\times T}\sum_{i=1}^{N}\sum_{j=1}^{T}{\left(y_{i,j}-{\hat{y}}_{i,j}\right)}^{2}\right)}^{1/2}\\ \; & MAE=\frac{1}{N\times T}\sum_{i=1}^{N}\sum_{j=1}^{T}\left|y_{i,j}-{\hat{y}}_{i,j}\right| \end{array}$$

### 4.3. Model Architecture and Hyperparameters

In this case study, we use three tasks to evaluate the prediction performance of the proposed deep learning model for both two cases. These tasks include the next 1st-hour traffic flow prediction (first task), the next 5th-hour traffic flow prediction (second task), and the next 10th-hour traffic flow prediction (third task). The next *n*th-hour traffic flow prediction refers to using the past 24 h traffic flow data to predict the traffic flow in the 24+n h. Table 2, Table 3 and Table 4 show the model architecture and hyperparameters used in this case study for three tasks. For these three tasks under two cases, we use the batch size of 100 and utilize the past 24 h traffic flow data to make predictions in each batch. We also use the filter size of 2 in the first task and use the filter size of 10 in the remaining two tasks. Moreover, the number of hidden nodes in the decoder network is 100.

### 4.4. Traffic Flow Prediction Results for the First Case

Figure 5 shows the traffic flow prediction results for three different tasks under the first case, where VMT refers to vehicle miles traveled, and VHT refers to vehicle hours traveled. From these three figures, we can observe that the proposed methodology can predict the traffic flow with high accuracy, as the true VMT and VHT are close to the predicted VMT and VHT. For example, for the 5th-hour prediction task, the predicted VMT is 1.260 when the true VMT is 1.219. For the 1st-hour prediction task, the predicted VHT is 0.337 when the true VHT is 0.325. To further demonstrate the performance of the proposed method, we compare the proposed method with existing methods reported in the literature, and these methods are listed in Table 5. In this table, the D-ConvoLSTM method refers to the decoder network with the convolutional LSTM network; and the D-Attention method refers to the decoder network with the multi-head attention mechanism; LSTM refers to the long short-term memory network; LASSO refers to the least absolute shrinkage and selection operator; ANN refers to the artificial neural network.

Table 6 compares the traffic flow prediction performance of the proposed method with methods listed in Table 5 in terms of RMSE and MAE. From this table, we can conclude that the proposed method can predict traffic flow with high accuracy and outperforms existing data-driven methods. For example, for the 1st-hour task, the RMSE of the VMT prediction for the proposed method is 0.032, and the RMSE of other data-driven methods ranges from 0.038 to 0.088. For the 5th-hour task, the RMSE of the VHT prediction for the proposed method is 0.128; however, the RMSE of LSTM is 0.145, and the RMSE of ANN is 0.245.

### 4.5. Traffic Flow Prediction Results for the Second Case

Figure 6 shows the traffic flow prediction results for three different tasks under the second case, where VMT refers to vehicle miles traveled, and VHT refers to vehicle hours traveled. From this figure, we can observe that the proposed methodology can predict the traffic flow with high accuracy as the true VMT and VHT are close to the predicted VMT and VHT. For example, for the 5th-hour prediction task, the predicted VMT is 1.085 when the true VMT is 1.082. For the 1st-hour prediction task, the predicted VHT is 2.110 when the true VHT is 2.138. Table 7 compares the traffic flow prediction performance of the proposed method with methods listed in Table 5 in terms of RMSE and MAE. From this table, we can conclude that the proposed method can predict traffic flow with high accuracy and outperforms existing data-driven methods. For example, for the 1st-hour task, the RMSE of the VMT prediction for the proposed method is 0.053, and the RMSE of other data-driven methods ranges from 0.055 to 0.091. For the 5th-hour task, the MAE of the VHT prediction for the proposed method is 0.093; however, the RMSE of LSTM is 0.129, and the RMSE of ANN is 0.175.

## 5. Conclusions and Future Work

In this study, a deep learning approach was proposed to predict traffic flow. In the proposed deep learning approach, a convolutional long short-term memory network was used to consider the correlation of the extracted high-dimensional features and the temporal correlation of traffic flow data in a unified manner. Moreover, a multi-head attention mechanism was implemented to use the most relevant portion of the traffic flow data to make predictions so that the prediction performance can be improved. A traffic flow dataset collected from the Caltrans Performance Measurement System (PeMS) database was used to demonstrate the effectiveness of the proposed method. Experimental results have shown that the proposed method can accurately predict the traffic flow with a minimum RMSE of 0.032 and outperforms the existing data-driven methods in terms of RMSE and MAE. Future work will be directed to use the convolutional LSTM network to make traffic flow predictions under more complicated environments and conditions.

## Figures and Tables

**Figure 1 sensors-22-07994-f001:**
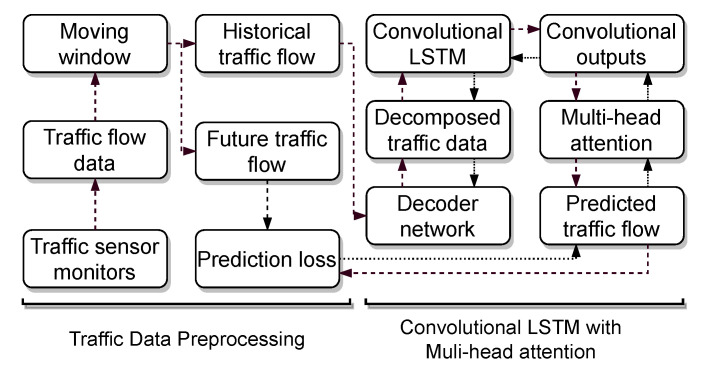
The framework of the convolutional LSTM with a multi-head attention mechanism for traffic flow prediction.

**Figure 2 sensors-22-07994-f002:**
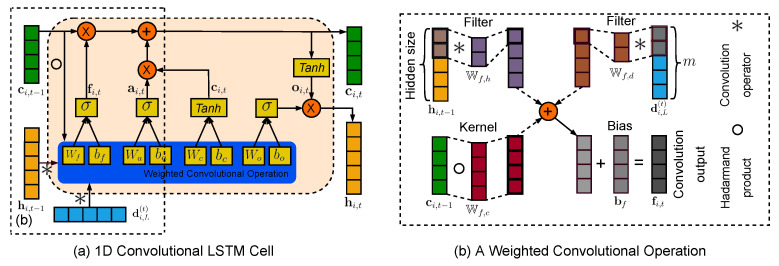
The framework of the one-dimensional convolutional LSTM cell with weighted convolutional operations, where (**a**) is the 1D convolutional LSTM cell and (**b**) gives an example of the weighted convolutional operation.

**Figure 3 sensors-22-07994-f003:**
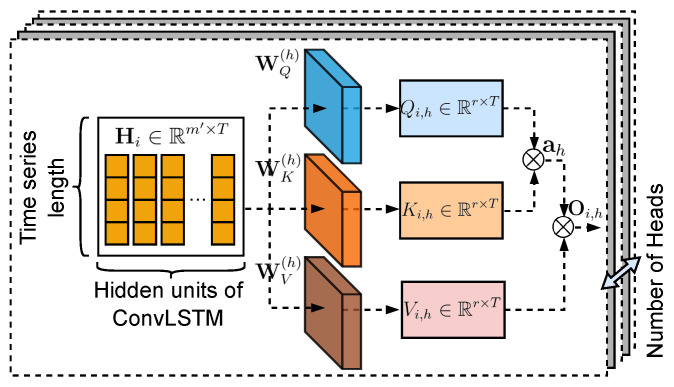
The framework of the multi-head attention model for traffic flow prediction.

**Figure 4 sensors-22-07994-f004:**
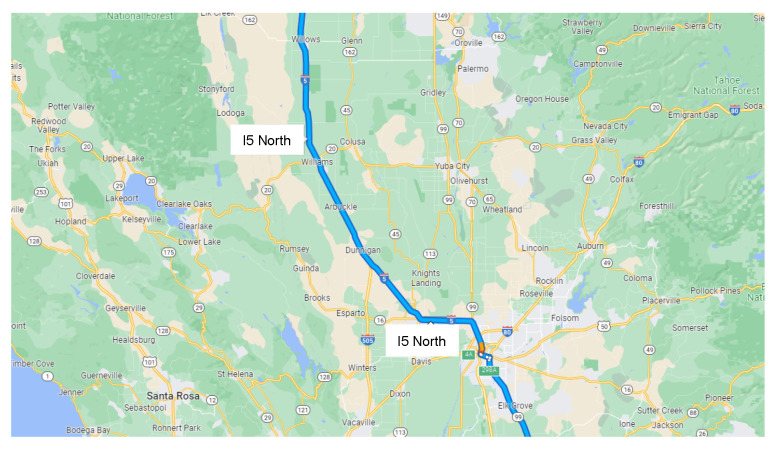
The post-mile ranges from 495.73 to 621.42 located at the freeway I5-North.

**Figure 5 sensors-22-07994-f005:**
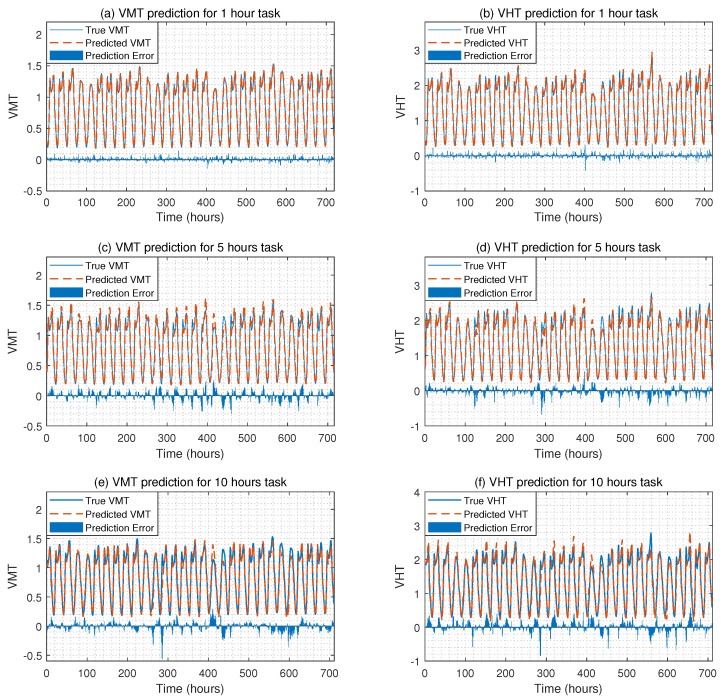
The VMT and VHT prediction results for three different tasks under the first case, where (**a**,**c**,**e**) show the VMT predictions for three tasks; and (**b**,**d**,**f**) show the VHT predictions for three tasks.

**Figure 6 sensors-22-07994-f006:**
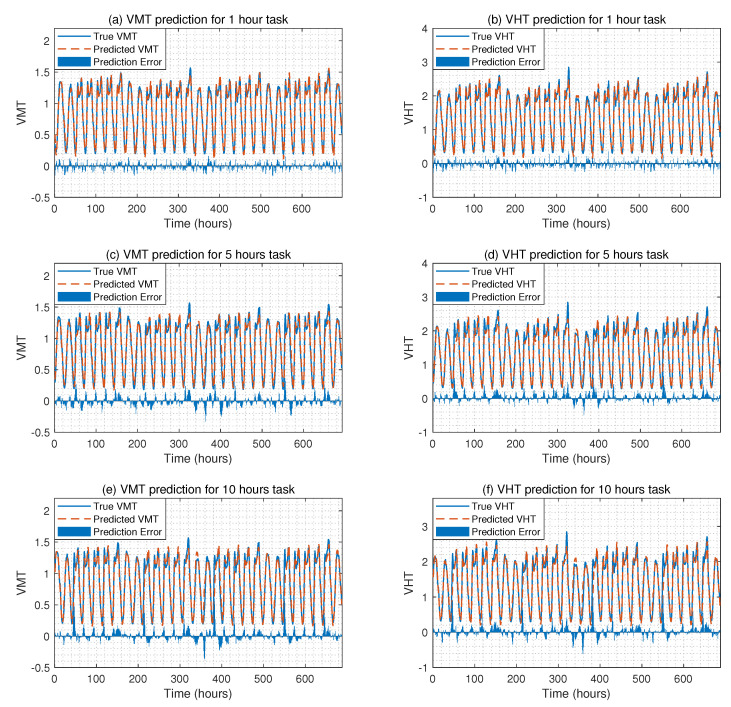
The VMT and VHT prediction results for three different tasks under the second case, where (**a**,**c**,**e**) show the VMT predictions for three tasks; and (**b**,**d**,**f**) show the VHT predictions for three tasks.

**Table 1 sensors-22-07994-t001:** The pseudo-code to train the proposed deep learning model for traffic flow predictions.

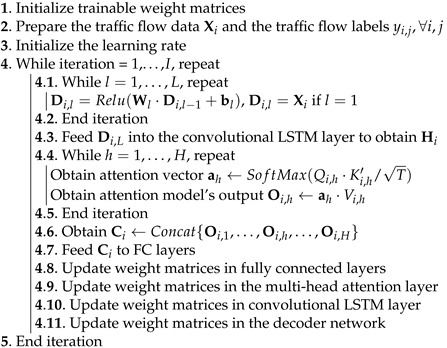

**Table 2 sensors-22-07994-t002:** The model architecture and hyperparameters used for the next 1st-hour traffic flow prediction task.

No. of Layers	Descriptions	Output Dimensions
1	Input layer	100 × 24 × 1
2	FC layer	100 × 24 × 100
3	Convolutional LSTM	100 × 24 × 99
4	Multi-head attention	100 × 24 × 99
5	Flatten layer	100 × 2376
6	Dense layer	100 × 1

**Table 3 sensors-22-07994-t003:** The model architecture and hyperparameters used for the next 5th-hour traffic flow prediction task.

No. of Layers	Descriptions	Output Dimensions
1	Input layer	100 × 24 × 1
2	FC layer	100 × 24 × 100
3	FC layer	100 × 24 × 100
4	FC layer	100 × 24 × 100
5	Convolutional LSTM	100 × 24 × 91
6	Multi-head attention	100 × 24 × 91
7	Flatten layer	100 × 2184
8	Dense layer	100 × 1

**Table 4 sensors-22-07994-t004:** The model architecture and hyperparameters used for the next 10th-hour traffic flow prediction task.

No. of Layers	Descriptions	Output Dimensions
1	Input layer	100 × 24 × 1
2	FC layer	100 × 24 × 100
3	FC layer	100 × 24 × 100
4	FC layer	100 × 24 × 100
5	FC layer	100 × 24 × 100
6	FC layer	100 × 24 × 100
7	Convolutional LSTM	100 × 24 × 91
8	Multi-head attention	100 × 24 × 91
9	Flatten layer	100 × 2184
10	Dense layer	100 × 1

**Table 5 sensors-22-07994-t005:** Symbols and descriptions of the proposed method and other methods for traffic flow predictions.

Method Symbol	Description
D-ConvLSTM	Decoder with convolutional LSTM
D-Attention	Decoder with multi-head attention
LSTM	Long short term memory network
LASSO	Regression with l1-norm regularization
ANN	Artificial neural network

**Table 6 sensors-22-07994-t006:** The traffic flow prediction errors in terms of RMSE and MAE for the proposed methods and other data-driven methods under the first case.

		1 h Task		5 h Task		10 h Task	
		VMT	VHT	VMT	VHT	VMT	VHT
RMSE	Proposed	0.032	0.066	0.080	0.128	0.084	0.167
	D-ConvLSTM	0.044	0.079	0.099	0.128	0.094	0.157
	D-Attention	0.043	0.086	0.105	0.179	0.113	0.199
	LSTM [30]	0.038	0.064	0.065	0.145	0.104	0.191
	LASSO [48]	0.088	0.141	0.142	0.242	0.141	0.240
	ANN [49]	0.054	0.103	0.137	0.245	0.138	0.241
MAE	Proposed	0.024	0.048	0.059	0.090	0.058	0.116
	D-ConvLSTM	0.034	0.058	0.072	0.097	0.064	0.115
	D-Attention	0.034	0.066	0.076	0.135	0.077	0.138
	LSTM [30]	0.029	0.045	0.046	0.107	0.064	0.130
	LASSO [48]	0.063	0.096	0.099	0.165	0.098	0.163
	ANN [49]	0.039	0.072	0.090	0.172	0.089	0.168

**Table 7 sensors-22-07994-t007:** The traffic flow prediction errors in terms of RMSE and MAE for the proposed methods and other data-driven methods under the second case.

		1 h Task		5 h Task		10 h Task	
		VMT	VHT	VMT	VHT	VMT	VHT
RMSE	Proposed	0.053	0.100	0.084	0.135	0.100	0.172
	D-ConvLSTM	0.088	0.153	0.094	0.157	0.118	0.184
	D-Attention	0.055	0.087	0.112	0.168	0.141	0.253
	LSTM [30]	0.055	0.106	0.113	0.187	0.119	0.225
	LASSO [48]	0.091	0.145	0.149	0.256	0.145	0.248
	ANN [49]	0.063	0.112	0.143	0.255	0.146	0.256
MAE	Proposed	0.042	0.078	0.062	0.093	0.064	0.109
	D-ConvLSTM	0.060	0.107	0.060	0.104	0.078	0.125
	D-Attention	0.043	0.107	0.075	0.123	0.104	0.187
	LSTM [30]	0.046	0.076	0.077	0.129	0.080	0.153
	LASSO [48]	0.063	0.099	0.102	0.175	0.100	0.168
	ANN [49]	0.044	0.079	0.096	0.175	0.099	0.179

## Data Availability

The dataset used in this study is available at https://pems.dot.ca.gov/ (accessed on 5 February 2022).

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
