# Peer review of "Convolutional Long-Short Term Memory Network with Multi-Head Attention Mechanism for Traffic Flow Prediction"

_sensors, 2022, doi:10.3390/s22207994_

Round 1

Reviewer 1 Report

- The mathematical formulation in section 3 is mainly from the literature and not a contribution of the paper, so please reduce drastically.

- Define the postmile for understanding by the wider readership. 

- Line 192, "To avoid loss of generality, both training and test data were standardized.", elaborate.

- Precisely define "the  next nth-hour traffic flow prediction"

- Provide the bland-altman plot for agreement measurement if suitable, or provide other metrics such as Forecasting Value (FV) or mean absolute error.. 

- There is comparison to related works with the same or other datasets.

- The table of abbreviations is missing.

Reviewer 2 Report

This paper proposes a deep learning approach for traffic flow prediction. This is an important topic and this paper is a solid study. However, there are still many concerns.

1. The authors should enhance their contributions as discussed in the end of the Introduction section in Page 2, since the multi-head attention layer and the convolutional LSTM layer are widely used in the literature.

2. The literature review is not comprehensive. The authors should add more discussion for the latest progresses of applying GNNs for traffic flow prediction.

[1] Jiang W, Luo J. Graph neural network for traffic forecasting: A survey[J]. Expert Systems with Applications, 2022: 117921.

[2] Ye J, Zhao J, Ye K, et al. How to build a graph-based deep learning architecture in traffic domain: A survey[J]. IEEE Transactions on Intelligent Transportation Systems, 2020.

3. The dataset and evaluation metric used in Section 4 is limited, when similar studies are using 2-3 different datasets and RMSE/MAE/MAPE metrics.

4. The resolution of Figure 4 in Page 8 should be improved.

5. Some small typos still exist. For example, "Thepseudo-codee" in the table 1 caption in Page 7.

Round 2

Reviewer 1 Report

The authors addressed the majority of my comments.

Reviewer 2 Report

Dear authors,

Thanks for revising and resubmitting the manuscript. The previous concerns are solved and no further comments.